# MultiContrievers: Analysis of Dense Retrieval Representations

**Seraphina Goldfarb-Tarrant**[♡♠*], **Pedro Rodriguez**[♢], **Jane Dwivedi-Yu**[♢], **Patrick Lewis**[♡]

♡ Cohere, ♠ University of Edinburgh, ♢ FAIR, Meta

{seraphina, patrick}@cohere.com
{par, janeyu}@meta.com

## Abstract

Dense retrievers compress source documents into (possibly lossy) vector representations, yet there is little analysis of what information is lost versus preserved, and how it affects downstream tasks. We conduct the first analysis of the information captured by dense retrievers compared to the language models they are based on (e.g., BERT versus Contriever). We use 25 MultiBert checkpoints as randomized initialisations to train **MultiContrievers**, a set of 25 contriever models. We test whether specific pieces of information—such as gender and occupation—can be extracted from contriever vectors of wikipedia-like documents. We measure this *extractability* via information theoretic probing. We then examine the relationship of extractability to performance and gender bias, as well as the sensitivity of these results to many random initialisations and data shuffles. We find that (1) contriever models have significantly increased extractability, but extractability usually correlates poorly with benchmark performance 2) gender bias is present, but is *not* caused by the contriever representations 3) there is high sensitivity to both random initialisation and to data shuffle, suggesting that future retrieval research should test across a wider spread of both.[1]

## 1 Introduction

Dense retrievers (Karpukhin et al., 2020; Izacard et al., 2022; Hofstätter et al., 2021) are a standard component of retrieval augmented Question Answering (QA) (Lewis et al., 2020a), and other retrieval systems such as fact-checking (Thorne et al., 2018), argumentation (Wachsmuth et al., 2018), and others. Despite their ubiquity, we lack an understanding of the information recoverable

---

* Work done while interning at FAIR, Meta.

[1]We release our 25 MultiContrievers (including intermediate checkpoints), all code, and all results, to facilitate further analysis. https://github.com/facebookresearch/multicontrievers-analysis

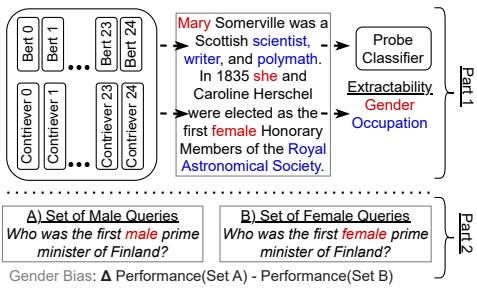

Figure 1: Part 1: We train 25 Contrievers from the 25 MultiBerts, and compare the information theoretic extractability of *gender* and *occupation* from each of their representations of documents. Part 2: We then compare these to metrics of performance and of gender bias to better understand the properties of dense retrievers.

from dense retriever representations, and how it affects retrieval system behaviour. This lack of analytical work is surprising. Retrievers are widespread, and are used in contexts that require trust: increasing factuality and decreasing hallucination (Shuster et al., 2021), and providing trust and transparency (Lewis et al., 2020b) via a source document that has provenance and can be examined. The information a representation retains from a source document constrains these abilities. Dense retrievers lossily encode input documents into N-dimensional representations, and by doing so necessarily emphasise some pieces of information over others. A biography of Mary Somerville will contain many details about her: her profession (astronomy and mathematics), her gender (female), her political influence (women's suffrage), her country of origin (Scotland) and others. Each of these features are relevant to different kinds of queries. Which ones will a given retriever represent most recoverably?

Some analysis of this type exists for Masked Language Models (MLMs) (§2.2), but there is no such analysis for retrievers, which optimise a contrastive loss. Contrastive training is a very different objective than MLM, based on (dis)similarity of paired samples. The choice of pair affects fea-

ture suppression – what is recoverable and what is not (Robinson et al., 2021). So we extend this previous analytical work into the retrieval domain, by training 25 **MultiContrievers** initialised from MultiBert checkpoints (Sellam et al., 2022). This is the first study that includes variability over a large number of retriever initialisations, with some surprising results from this alone. We use information theoretic probing, also known as minimum description length (MDL) probing (Voita and Titov, 2020), to measure the information in MultiContriever representations. We evaluate the models on 14 retrieval datasets from the BEIR benchmark (Thakur et al., 2021). We test how well retrievers preserve information in a document, like gender and occupation, which we refer to as *features*. We adapt the existing datasets to better test for knowledge of these, by creating a new manually annotated gender subset of Natural Questions, **NQ-gender**. We ultimately test if gender information is predictive of gender bias, as it was in previous MLM work (§2.2).We address the following four research questions:

**Q1** *To what extent do retrievers preserve information like gender and occupation in an encoded document?* (§4.1)

For both MultiBerts and MultiContrievers, gender is more extractable than occupation, which can cause a model to rely on gender heuristics (a source of gender bias). But there are noticeable differences in the models. *Both* features are more extractable in MultiContrievers than MultiBerts, but there is a lower *ratio* (less difference) between gender and occupation. This indicates MultiContrievers are less *likely* to rely less on gender heuristics (Lovering et al., 2021), but still might.

**Q2** H*ow sensitive is this to random initialisation and data shuffle?* (§4.2)

In MultiBerts, extractability is very sensitive to random initialisation and shuffle, in MultiContrievers it is not. MultiContrievers have a much smaller variance between the 25 seeds, suggesting a regularising effect. However, MultiContriever *performance* is surprisingly sensitive to both random initialisation and to data shuffle. MultiContrievers have a very wide range of performance on BEIR benchmarks, despite identical loss curves. But it is not easy to select a 'best' model, since the best and worst model is not consistent across datasets - the ranking of each model can change, sometimes drastically.

**Q3** *Do differences in this information correlate with performance on retrieval benchmarks?* (§4.3)

On partitions of examples that ostensibly require gender information (NQ-gender), we show that gender extractability is highly correlated with retrieval performance. However, overall retrieval performance on benchmarks like BEIR is poorly correlated with extractability. This suggests that while some benchmark examples do reward models for preserving gender information, most examples do not require that, so the benchmark as a whole does not require that capability.

**Q4** *Is gender information in retrievers predictive of their gender bias?* (§4.4)

Despite the evidence that extractability of gender information is helpful to a model, it is *not* the cause of gender bias in the NQ-gender dataset. When we do a causal analysis by removing gender from MultiContriever representations, gender bias persists, suggesting that the source of bias is in the queries or corpus.

Our contributions are: **1)** the first information theoretic analysis of dense retrievers, **2)** an analysis of variability in performance and social bias across random retriever seeds, **3)** the first causal analysis of sources of social bias in dense retrievers, **4) NQ-gender**, an annotated subset of Natural Questions for queries that constrain gender, and **5)** a suite of 25 **MultiContrievers** for use in future work, with all training and evaluation code.

## 2 Background and Related Work

The below covers dense retrievers, information theoretic probing for extractability, and what extractability can tell us about model behaviour.

### 2.1 What is a retriever?

Retrievers take an input query and return relevance scores for documents from a corpus. We encode documents $D$ and queries $Q$ separately by the same model $f_\theta$. Given a query $q_i$ and document $d_i$, relevance is the dot product between the document and query representations.

$$s(d_i, q_i) = f_\theta(q_i) \cdot f_\theta(d_i) \qquad (1)$$

Training $f_\theta$ is a challenge. Language models like BERT (Devlin et al., 2019), are not good retrievers out-of-the-box, but retrieval training resources are limited and labour intensive to create, since they involve matching candidate documents to a query from a corpus of potentially millions. So retrievers are either trained on one of the

few corpora available, such as Natural Questions (NQ) (Kwiatkowski et al., 2019) or MS MARCO (Campos et al., 2016) as supervision (Hofstätter et al., 2021; Karpukhin et al., 2020), or on a self-supervised proxy for the retrieval task (Izacard et al., 2022). Both approaches result in a domain shift between training and later inference, making retrieval a *generalisation task*. This motivates our analysis, as Lovering et al. (2021)'s work shows that information theoretic probing is predictive of where a model would generalise and where it relies on simple heuristics and dataset artifacts.

In this work, we focus on the self-supervised Contriever (Izacard et al., 2022), initialised from a BERT model and then fine-tuned with a contrastive objective.[2] For this objective, all documents in a large corpus are broken into chunks, where chunks from the same document are positive pairs and chunks from different documents are negative pairs. This is a loose proxy for 'relevance' in the retrieval sense, so we are interested in what information this objective encourages contriever to emphasise, what to retain, what to lose, and what this means for the eventual retrieval task.

## 2.2 What is Information Theoretic (MDL) probing?

Diagnostic classifiers, or **probes**, are a powerful tool for determining what information is in a model representation (Belinkov and Glass, 2019). Let $DS = \{(d_i, y_i), ..., d_n, y_n)\}$ be a dataset, where $d$ is a document (e.g. a chunk of a Wikipedia biography about Mary Somerville) and $y$ is a label from a set of $k$ discrete labels $y_i \in Y$, $Y = \{1, ...k\}$ for some information in that document (e.g. *mathematics, astronomy* if probing for occupation).

In a probing task, we want to measure how well $f_\theta(d_i)$ encodes $y_i$, for all $d_{1:n}$, $y_{1:n}$. We use Minimum Description Length (MDL) probing (Voita and Titov, 2020), or information theoretic probing, in our experiments. This measures **extractability** of $Y$ via compression of information $y_{1:n}$ from $f_\theta(d_{i:n})$ via the ratio of uniform codelength to online codelength.

$$Compression = \frac{L_{uniform}}{L_{online}} \quad (2)$$

---

[2]We choose Contriever for societal relevance of our results, as it has two orders of magnitude more monthly downloads than other popular models: https://huggingface.co/facebook/contriever.

where $L_{uniform}(y_{1:n}|f_\theta(d_{i:n}) = n \log_2 k$ and $L_{online}$ is calculated by training the probe on increasing subsets of the dataset, and thus measures quality of the probe relative to the number of training examples. Better performance with less examples will result in a shorter online codelength, and a higher compression, showing that $Y$ is more extractable from $f_\theta(d_{i:n})$.

In this work, we probe for binary *gender*, where $Y = \{m, f\}$ and *occupation*, where $Y = \{lawyer, doctor, ...\}$

**Extractability**, as measured by MDL probing, is predictive of *shortcutting*; when a model relies on a heuristic feature to solve a task, which has sufficient correlation with the actual task to have high accuracy on the training set, but is not the true task (Geirhos et al., 2020). Shortcutting causes failure to generalise; a heuristic that worked on the training set due to a spurious correlation will not work after a distributional shift: e.g. relying on the word 'not' to predict negation may work for one dataset but not all (Gururangan et al., 2018). Lovering et al. (2021) look at linguistic information in MLM representations (such as subject verb agreement) which is necessary for the task of grammaticality judgments, and find that spurious features are relied on if they are very extractable. This is of particular interest to retrievers, which depend on generalisation, but which are also contrastively trained, which can encourage shortcutting (Robinson et al., 2021).

Shortcutting is also often the cause of social biases. Orgad et al. (2022) find that extractability of gender in language models is predictive of gender bias in coreference resolution and biography classification. So when some information, such as gender, is more extractable than other information, such as anaphora resolution, the model is risk of using gender as a heuristic, if the data supports this usage. And thus of both failing to generalise and of propagating biases. For instance, for the case of Mary Somerville, if gender is easier for a model to extract than profession, then a model might have actually learnt to identify mathematicians via *male*, instead of via *maths* (the true relationship), since it is both easier to learn and the error penalty on that is small, as there are not many female mathematicians.

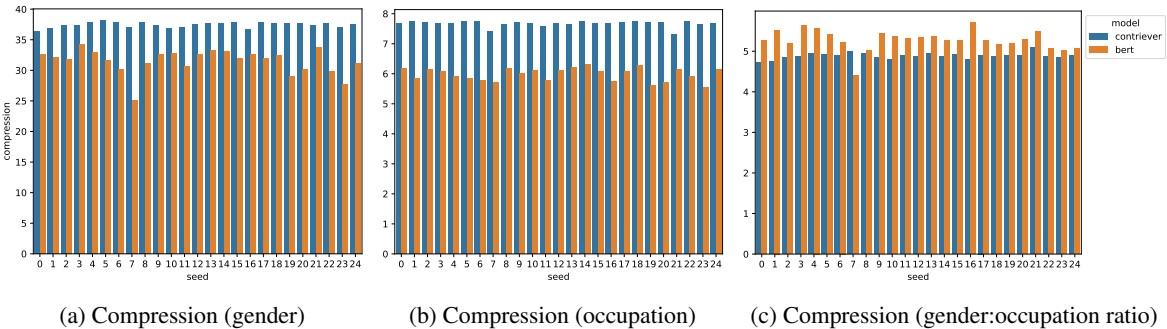

(a) Compression (gender)      (b) Compression (occupation)      (c) Compression (gender:occupation ratio)

Figure 2: Bert and Contriever compression for gender and occupation over all seeds. Y-axes have different scales (gender is much larger); higher numbers mean more extractability and more regular representations. Contriever has more uniform compression across seeds, and a lower ratio of gender:occupation, which means less shortcutting.

## 3 Methodology

We analyse the relationship between information in different model representations, and their performance & fairness. This requires at minimum a model, a probing dataset (with labels for information we want to probe for), and a performance dataset. We need some of the performance dataset to have gender metadata to calculate performance difference across gender demographics (Fig 1) also called gender bias, or more precisely, *allocational fairness*.

### 3.1 Models

For the majority of our experiments, we compare our 25 MultiContriever models to the 25 Multi-Berts models (Sellam et al., 2022). We access the MultiBerts via huggingface[3] and train the contrievers via modifying the repository released in Izacard et al. (2022). We use the same contrastive training data as Izacard et al. (2022), to maximise comparability. This comprises a 50/50 mix of Wikipedia and CCNet from 2019. As a result, five of the fourteen performance datasets involve temporal generalisation, since they postdate both the MultiContriever and the MultiBert training data. This most obviously affects the TREC-COVID dataset (QA), though also four additional datasets: Touché-2020 (argumentation), SCIDOCS (citation prediction), and Climate-FEVER and Sci-fact (fact-checking). Further details on contriever training and infrastructure are in Appendix G.

We train 25 random seeds as both generalisation and bias vary greatly by random seed initialisation (McCoy et al., 2020). MultiContrievers have no new parameters, so the random seed affects only their data shuffle. The MultiBerts each

have a different random seed for both weight initialisation and data shuffle.

### 3.2 Probing Datasets

We verify that results are not dataset specific, or the result of dataset artifacts, by using two probing datasets. First the BiasinBios dataset (De-Arteaga et al., 2019), which contains biographies from the web annotated with labels of the subject's binary gender (male, female) and biography topic (lawyer, journalist, etc). We also use the Wikipedia dataset from md_gender (Dinan et al., 2020), which contains Wikipedia pages about people, annotated with binary gender labels.[4] For gender labels, BiasinBios is close to balanced, with 55% male and 45% female labels, but Wikipedia is very imbalanced, with 85% male and 15% female. For topic labels, BiasinBios has a long-tail zipfian distribution over 28 professions, with professor and physician together as a third of examples and rapper and personal trainer as 0.7%. Examples from both datasets can be found in Appendix A.

To verify the quality of each dataset's labels, we manually annotated 20 random samples and compared to gold labels. BiasinBios agreement with our labels was 100%, and Wikipedia's was 88%.[5] We focus on the higher quality BiasinBios dataset for most of our graphs and analysis, though we replicate all experiments on Wikipedia.

---

[3]e.g. https://huggingface.co/google/MultiBerts-seed_[SEED]

[4]This dataset does contain non-binary labels, but there are few (0.003%, or ~180 examples out of 6 million). Uniform codelength ($dataset\_size * log2(num\_classes)$) affects information theoretic probing; additional class with very few examples can significantly affect results. This dataset was also noisier, making small data subsets less trustworthy.

[5]We investigated other md_gender datasets in the hope of replicating these results on a different domain such as dialogue (e.g. Wizard of Wikipedia), but found the labels to be of insufficiently high agreement to use.

## 3.3 Evaluation Datasets and Metrics

We evaluate on the BEIR benchmark, which covers retrieval for seven different tasks (fact-checking, citation prediction, duplicate question retrieval, argument retrieval, question answering, bio-medical information retrieval, and entity retrieval).[6]. We initially analysed all standard metrics used in BEIR and TREC (e.g. NDCG, Recall, MAP, MRR, @10 and @100). We observed similar trends across all metrics, somewhat to our surprise, since many retrieval papers focus on the superiority of a particular metric (Wang et al., 2013). We thus predominantly report NDCG@10, but more metrics (NDCG@100, and Recall@100) are included in Appendix F.

For allocational fairness evaluation, we create **NQ-gender**, a subset of Natural Questions (NQ) about entities, annotated with male, female, and neutral (no gender). Further details on annotation in Appendix B. We measure allocational fairness as the difference between the female and male query performance. We use the neutral/no gender entity queries as a control to make sure the system performs normally on this type of query.

## 4 Results

We address our four research questions: how does extractability change (Q1), how sensitive are retrievers to random initialisation (Q2), do changes in extractability correlate with performance (Q3), and is it predictive of allocational bias (Q4).

### 4.1 Q1: Information Extractability

Both gender (Fig 2a) and occupation (Fig 2b) are more extractable in MultiContrievers than MultiBerts. Gender compression ranges for MultiContrievers are 4-12 points higher, or a 9-47% increase (depending on seed initialisation), than the corresponding MultiBerts. Occupation compression ranges are 1.7-2.12 points higher for MultiContrievers; as the overall compression is much lower this is a 19-38% increase over MultiBerts. Both graphs also show a regularisation effect; MultiBerts have a large range of compression across random seeds, whereas MultiContrievers have similar values.

Figure 2c shows that though MultiContrievers have higher extractability for gender and occupa-

---

[6]The BEIR benchmark itself contains two additional tasks, tweet retrieval, and news retrieval, but these datasets are not publicly available.

tion, the ratio between them decreases. So while MultiContrievers do represent gender far more strongly than occupation, this effect is lessened vs. MultiBerts, which means they should be slightly less likely to shortcut based on gender.

## 4.2 Q2: Sensitivity to Random Initialisation

We analysed the distribution of performance by dataset for 24 seeds, as both generalisation and fairness are sensitive to initialisation in MLMs (Sellam et al., 2022).[7] Figure 5 shows this data, broken out by dataset, with a dashed line at previous reference performance (Izacard et al., 2022).

A few things are notable: first, **there is a large range of benchmark performance across seeds with for identical contrastive losses.** During training, MultiContrievers converge to the same accuracy (Appendix G) and (usually) have the same aggregate BEIR performance reported in Izacard et al. (2022). However, the range of scores per dataset is often quite large, and for some datasets the original reference Contriever is in the tail of the distribution: e.g in Climate-Fever (row 1 column 2) it performs *much* worse than all 24 models. It is also worse than almost all models for Fiqa and Arguana.[8] Nothing changed between the different MultiContrievers except the random seed for MultiBert initialisation, and the random seed for the data shuffle for contrastive fine-tuning.[9]

Second, **the potential increase in performance across random seeds can exceed the increase in performance from training on supervised data (e.g. MSMARCO).** We see this effect for half the datasets in BEIR. The higher performing seeds surpass the performance on *all* supervised models from Thakur et al. (2021)[10] on

---

[7]Seed 13 (ominously) is excluded from our analysis because of extreme outlier behaviour, which was not reported in (Sellam et al., 2022). We investigated this behaviour, and it is fascinating, but orthogonal to this work, so we have excluded the seed from all analysis. Our investigation can be found in Appendix D and should be of interest to researchers investigating properties of good representations (e.g. anisotropy) and of random initialisations.

[8]For Fiqa 19 models are up to 2.5 points better, for Arguana 20 models are up to 6.3 points better.

[9]There are a few small differences between the *original* released BERT, which Contriever was trained on, and the MultiBerts, which we trained on, detailed in Sellam et al. (2022). But not between our 25 MultiBerts.

[10]The BEIR benchmark reports performance on all datasets for four dense retrieval systems—DPR(Karpukhin et al., 2020), ANCE (Xiong et al., 2021), TAS-B (Hofstätter et al., 2021), and GENQ (their own system)—which all use supervision of some kind. DPR uses NQ and Trivia QA, as well as two others, ANCE, GENQ, and TAS-B all use MS-

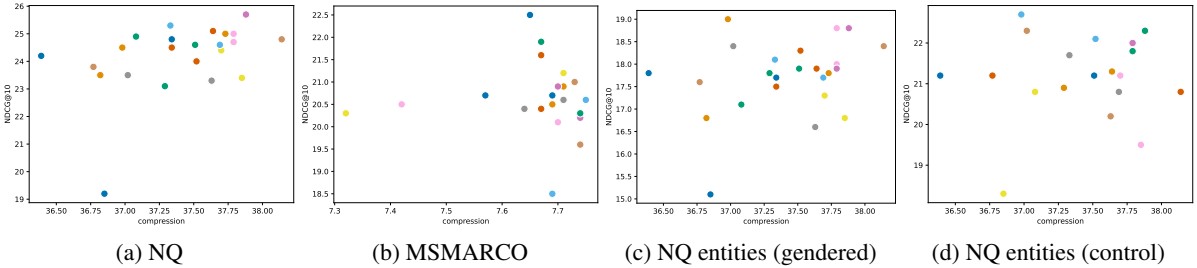

(a) NQ  (b) MSMARCO  (c) NQ entities (gendered)  (d) NQ entities (control)

Figure 3: Scatterplots of the correlation between x-axis compression (ratio of uniform to online codelength) and y-axis performance (NDCG@10), for different datasets (NQ, MSMARCO) at left and entity subsets of NQ at right. Colours are different seeds, and are held constant across graphs.

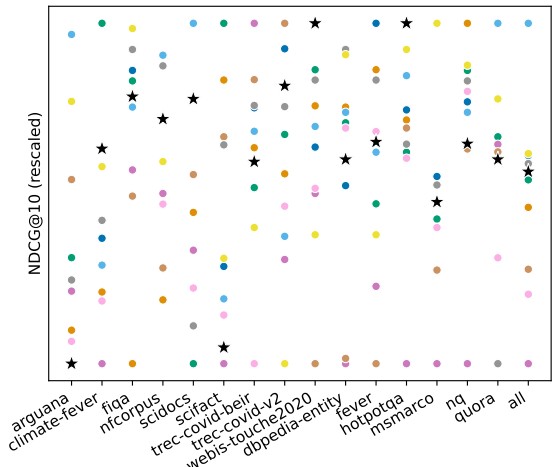

Figure 4: Ranking of best performing seed per dataset (one colour per seed). For legibility, NDCG@10 values are scaled, and all seeds with middle performance are not pictured (10 included). One seed is arbitrarily given a star marker to aid visual interpretation.

three datasets (Fever, Scifact, and Scidocs) and surpass all but one model (TAS-B) on Climate-fever. These datasets are the fact-checking and citation prediction datasets in the benchmark, suggesting that even under mild task shifts from supervision data (which is always QA), random initialisation can have a greater effect than supervision. This effect exists across diverse non-QA tasks; for four additional datasets the best random seeds are better than all but one supervised model: this is true for Arguana and Touché (argumentation), HotpotQA (multihop QA), and Quora (duplicate question retrieval).

Third, **the best and worst model across the BEIR benchmark datasets is not consistent** (Figure 4); not only is the range large across seeds but the ranking of each seed is very variable. The best model on average, seed 24, is top-ranked on only *one* dataset, and the second-best average

MARCO. Note that the original Contriever underperformed these other models until supervision was added.

model, seed 2, is best on *no* individual datasets. The best or worst model on any given dataset is almost always the best or worst on *only* that dataset and none of the other 14. Sometimes, the best model on one dataset is worst on another, e.g. seed 4 is best on NQ and worst on FiQA, seed 5 is best on Scifact and worst on Scidocs.[11] Even seed 10, which is the only model that is worst on more than 2 datasets (it is worst on 6) is still best on TREC-Covid.[12]

Our results show that there is no single best retriever, which both supports the motivation of the BEIR benchmark (to give a more well rounded view on retriever performance via a combination of diverse datasets) and shows the need for more analysis into random initialisation and shuffle.

As an addendum, we note that Sellam et al. (2022) did extensive experiments with both random initialisation and data shuffle, and found initialisation to matter more. We did our own experiments to this effect where we trained five MultiContrievers from the same MultiBert initialisation with different data shuffles, from the best, worst, and middle performing seeds. This additional analysis is in Appendix C.

### 4.3 Q3: Correlation between Extractability & Performance

We tested for correlations across all datasets and common metrics, and present a selection here (Fig 3). Neither NQ (Fig 3a) nor MSMARCO (Fig 3b) correlate with compression metrics. NQ and MSMARCO are the most widely used of the BEIR benchmark datasets, and we hypothesised them to be most likely to correlate. Both are

---

[11]This best-worst flip exists for seeds 8, 18, and 23 also.

[12]This is to be taken with a grain of salt - that dataset is interesting for generalisation (as these models are trained on only pre-Covid data), but it is only 50 datapoints. We note also that analysis on seed 13 revealed that seed 10 was also unusual, that analysis can also be found in D.

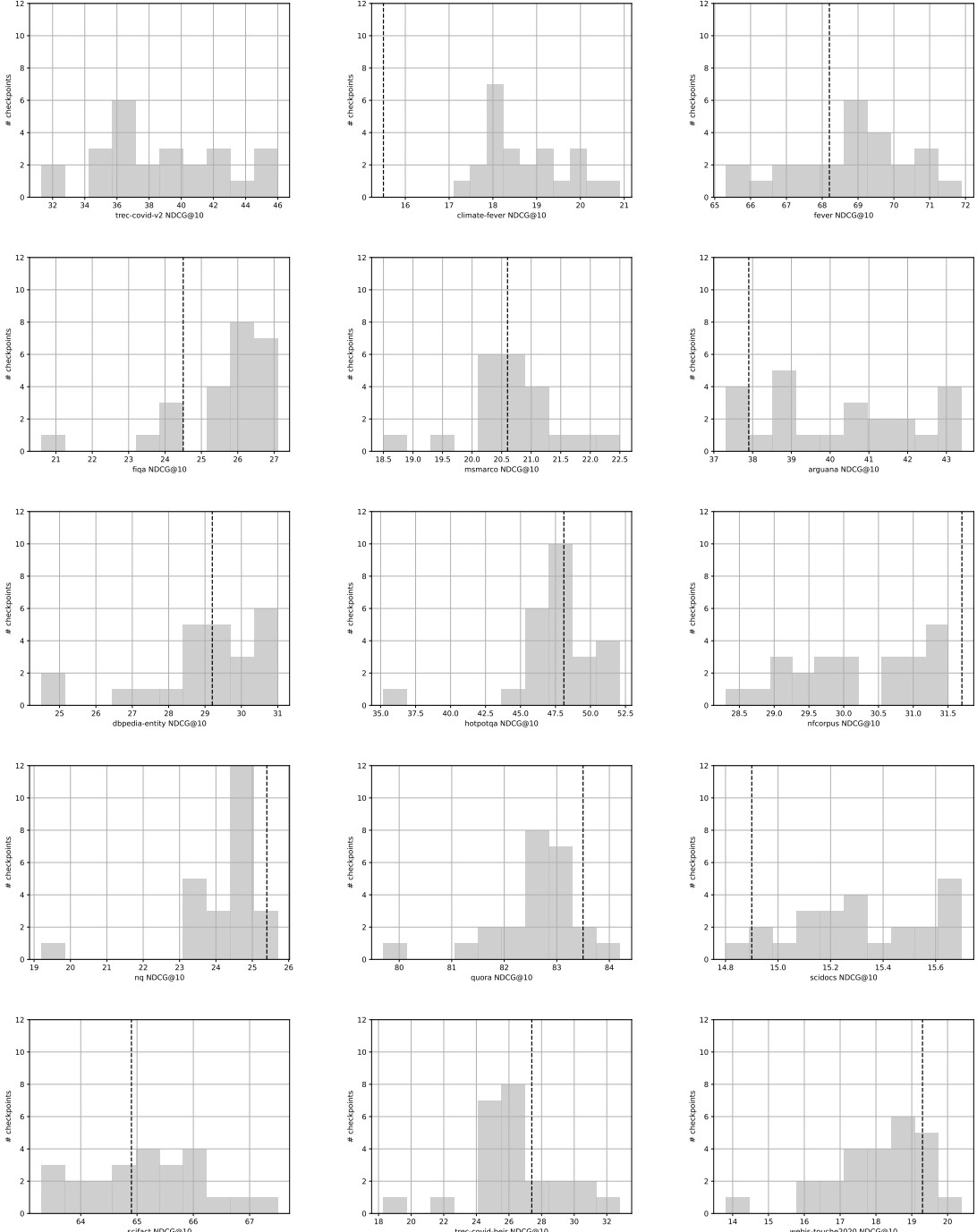

Figure 5: Distribution of performance (NDCG@10) for the 24 MultiContrievers, per BEIR dataset, performance on x-axis, number of models with that value on y-axis. Dashed line indicates reference performance from previous work. While for some datasets the reference performance sits at or near the mean of the MultiContriever distribution, for some the reference performance is an outlier. There is sufficiently high variance that performance improvements from random seed can exceed those from continuing to train on supervised fine-tuning data.

search engine queries (from Google and Bing, respectively) and contain queries that require occupation-type information (*what is cabaret music?*, MSMARCO) and that require gender information (*who is the first foreign born first lady?*, NQ). However, as the dispersed points on the scatterplots show (Figures 3a, 3b), neither piece of

information correlates to performance on either dataset. NQ and MSMARCO are representative; we include plots for all datasets in Appendix E.

This result was somewhat surprising; since the contriever training both regularises and increases extractability of gender and occupation, we might expect this to be important for the task. But per-

haps it is relevant for *only* the contrastive objective, and not for the retrieval benchmark. Alternatively, it is possible that this information is important, but only up to some threshold that MultiContriever models exceed. Finally, it's possible that this information doesn't matter for most queries in these datasets, and so there is some correlation but it is lost, as these datasets are extremely large. This is somewhat supported by the exception cases with correlations being smaller, more curated datasets (E), and so we investigated this as the most tractable to implement.

Our NQ-gender subset of gendered queries (§3.3) does show a strong correlation between gender extractability and performance (Fig 3c). And the NQ-gender subset of neutral non-gendered queries shows no correlation (Fig 3d). So we find that if we isolate to a topical dataset, as e here, extractability *is* predictive of performance, it just isn't over a large diverse dataset.

We strengthen this analysis, testing whether gender information is *necessary*, rather than simply correlated. We use Iterative Nullspace Projection (INLP) (Ravfogel et al., 2020) to remove gender information from MultiContriever representations. INLP learns a projection matrix $W$ onto the nullspace of a gender classifier, which we apply *before* computing relevance scores between corpus and query. So with INLP, the previous Equation 1 becomes:

$$s(d_i, q_i) = \mathbf{W}f_\theta(q_i) \cdot \mathbf{W}f_\theta(d_i) \qquad (3)$$

Then we calculate performance of retrieval with these genderless representations. No drop in performance on the gendered query set with INLP would mean extractable gender information was not necessary. A drop in performance on *both* gender and control queries would support the 'minimum threshold' explanation, or mean that the representation was sufficiently degraded by the removal of gender that other functions were harmed.

Gender information post-INLP drops to 1.4 (nearly none, as 1 is no compression over uniform, Eq 2). Performance on non-gendered entity queries is unaffected, but performance on gendered entity questions drops significantly (5 points) (Fig 6a). From these two experiments we conclude that the increased information extractability *was* useful for answering specific questions that require that information. But most queries in the available benchmarks simply don't require that information to answer them.

 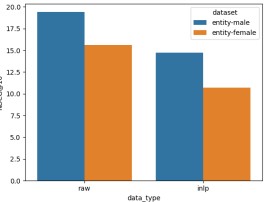

(a) NDCG@10 on the neutral vs. gendered NQ entity subsets. Representations are raw (blue) vs. INLP (orange) with gender removed. INLP performance degrades on *only* gender constrained queries: gender is used in those queries, but is not in the control.

(b) Difference in performance between male (blue) and female (orange) entity queries, for raw (left) and INLP (right). The performance gap is constant even when gender is removed via INLP, remains; so the bias is not due to gender in the representations.

Figure 6: INLP experiments

## 4.4 Q4: Gender Extractability and Allocational Gender Bias

Orgad et al. (2022) found gender extractability in representations to be predictive of allocational gender bias for classification tasks; when gender information was reduced or removed, bias also reduced.[13] We found that gender information is *used* (§4.3) so now we ask: is it predictive of gender bias? At least for our dataset, it is not (Fig 6b. This graph shows that there is allocational bias between the female and male queries, and also that the bias remains *after* we remove gender via INLP. *All* performance drops, as we saw for the gendered entities in §4.3. But performance drops by equivalent amounts for female and male entities. These results diverge from what we expected based on the findings of Orgad et al. (2022) for MLMs, who found gender in representations did matter. Our findings suggest that in this case the gender bias comes from the retrieval corpus or the queries, or from a combination. The corpus could have lower quality or less informative articles about female entities (as was found for Wikipedia by Sun and Peng (2021)), or queries about women could be structurally harder in some way.

## 5 Discussion, Future Work, Conclusion

We trained a suite of 25 **MultiContrievers**, analysed their performance on the BEIR benchmark, probed them for gender and occupation information, and removed gender information from representations to analyse gender bias.

We showed performance to be extremely

---

[13]Orgad et al. (2022) use a lexical method to remove gender, but we chose INLP as a more elegant, extensible solution. We replicated their paper with INLP, showing equivalence.

variable by random seed initialisation, as was the performance ranking of different random initialisations across datasets, despite equal losses during training. Best seed performances often exceed the performance of more complex dense retrievers that use explicit supervision. Future analysis of retriever loss basins to look for differing generalisation strategies could be valuable (Juneja et al., 2023). Our results show that a better understanding of initialisations may be more valuable than developing new models. Our work also highlights the usefulness of metadata enriched datasets for analysis, and we were limited by what was available. Future work could create these datasets and then probe for additional targeted information to learn more about retrievers. This would also enable analysis of demographic biases beyond binary gender.

Gender and occupation extractability was not predictive of performance except in subsets of queries that require gender information. Though both gender and occupation increase in Multi-Contrievers, the ratio between them decreases, so MultiContrievers should be less likely to shortcut based on gender compared to MultiBerts. We established that the gender bias that we found was not caused by the representations, as it persists when gender is removed. Future work should test in a pipeline is best to correct bias, and how various parts interact. This work also shows the utility of information removal (INLP, others) for causality and interpretability, rather than just debiasing. More availability of test sets for shortcutting could increase the scope of these preliminary results.

Finally, we have analysed only the retriever component of a retrieval system. In an eventual retrieval augmented generation task, the retrieval representation will have to compete with language model priors. The generation will be a composition between unconditionally probable text, and text attested by the retrieved data. Future work could investigate the role of information extractability in the full system, and how this bears on vital questions like hallucination in retrieval augmented generation. We have done the first information theoretic analysis of retrieval systems, and the first causal analysis of the reasons for allocational gender bias in retrievers. We release our code and resources for the community to expand and continue this line of enquiry. This is particularly important in the current generative NLP landscape, which is increasingly reliant on retrievers and where understanding of models lags so far behind development.

# 6 Limitations

This work is limited by analysing only one architecture of dense retriever; we chose to experiment instead with random initialisations and shuffles rather than different architectures, so we focused on only the most popular one. So these results may not generalise to all retriever architectures. Our analysis covered only English, and there is work that shows that gender is encoded in a more complex way in other languages (Gonen et al., 2022). INLP, the method we used for causal analysis, is linear, so it might not even work beyond English, though there are recent non-linear extensions of it (Iskander et al., 2023) that could be used in future work.

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

## A  Probing Datasets

We rely on two datasets. The first is BiasinBios (De-Arteaga et al., 2019), which is a dataset of web biographies labelled with binary gender, and biography profession. We use De-Arteaga et al. (2019)'s train/dev/test splits of 65:10:25, yielding 255,710 train 39,369 dev, and 98,344 test datapoints. Second is the Wikipedia slice of the `md_gender` dataset (Dinan et al., 2020). This has only labels for gender, which we restrict to be binary since non-binary gender is so small and would adversely affect this analysis. We filter out texts below 10 words (words, not tokens) leaving a dataset of size 10,681,700, split 65:10:25 into 6,943,105 train, 934,649 dev, 2,803,946 test. For practical reasons, we shard it to 9 shards (650,000 train examples each) and then check the results on each shard. All shards behaved consistently. As noted in the text, BiasinBios is nearly balanced with regard to gender labels, but Wikipedia is severely imbalanced.

For both datasets, we use the train set for probing, and the test set for measuring accuracy on the final probe. We investigated using other datasets, but none were of sufficient quality that they were usable. We tested usability very simply: each of the authors labelled a different random sample of 20 examples by hand, and we measured accuracy of dataset labels against our labels, and only took datasets with over 80% accuracy, since our probing task is sensitive to errors in labelling. No other subsets of md_gender nor external datasets that we surveyed passed this bar. We didn't multiply annotate as we found no examples to be at ambiguous.

## B  Annotation of NQ gender subset

To do our experiments we create a subset of Natural Questions, **NQ-gender**.

We subsample Natural Questions to entity queries by filtering automatically for queries containing any of `who, whose, whom, person, name`. We similarly filter this set into gendered entity queries by using a modified subset of gender terms from Bolukbasi et al. (2016). From this we get a set of queries that is just about entities *Who was the first prime minister of Finland?*, and gendered entities (a female query is *Who was the first female prime minister of Finland?* and a male query is *Who was the first male prime minister of Finland?*).

This automatic process is low precision/high

recall. It captures queries with gendered terms in prepositional phrases, (`Who starred in O Brother Where Art Thou?`) which are common false positives in QA datasets, as they are not about brothers. So we manually filter these results by annotating with two criteria: gender of the subject (male, female, or neutral/none (in cases where the gender term was actually in a title or other prepositional phrase as in the example), and a binary tag with whether the query actually *constrains* the gender of the answer. This second annotation is somewhat subtle, but very important. For example, in our dataset there is the query `Who was the actress that played Bee`, which contains a gendered word (actress) but it is not necessary to answer the question; all actors that played Bee are female, and the question could be as easily answered in the form `Who played Bee?`. Whereas in another example query, `Who plays the sister in Home Alone 3?` the query does constrain the gender of the answer. We annotated 816 queries with both of these attributes, of which 51% have a gender constraint, with a gender breakdown of 59% female and 41% male.

We do this annotation ourselves (two of the authors), and we throw out examples that we don't agree on. We are not a representative sample of people (we are all NLP researchers after all) but we consider this lack of diversity to be acceptable since we are not making subjective judgments but are just providing metadata labels.

It is also worth mentioning that two very different types of gender bias in retriever works do create artifacts also, but they are unsuitable for our type of analysis for the following reasons. Rekabsaz and Schedl (2020) and Klasnja et al. (2022) release subsets of MSMARCO, which we did examine and use in initial tests early in this work. Those works define bias very differently, as the genderedness of retrieved documents based on lexical terms, making the implicit normative statement that lack of bias means equal representation of male and female documents in non-gendered queries. This is essentially an independence assertion from fairness literature (Barocas et al., 2019). This is quite different to our approach, which looks at performance disparity between queries that require male and female gender information to answer. Our approach has more immediate practical utility for a real world retriever,

and also ties in to the work on information theory by restricting to queries that require gender information. So the lexical document based approach cannot be adapted for our purpose.

## C  Data Shuffle Experiments

We wanted to answer the question of *If you begin from a worse random initialisation, can you fix it via data shuffle?*. This is of significant practical utility to researchers, who often cannot retrain an existing model from scratch before adapting it to their purpose. Figure 7 shows the best, worst, and a middle performing seed with five additional different data shuffles, and the variance in performance over the datasets. We can see that the worst performing seed is characterised by high variability overall, and the best seed by low variability. So the overall picture is that, on average, the different initialisations determine the quality of the retriever more than the data shuffle. This is in agreement with the findings of Sellam et al. (2022) for MLMs. However, variability is sufficiently high enough that you could get lucky and get the best performance from varying the shuffle, if that is the option available. It would be valuable to extend these to explicit generalisation tasks and interpretabilty challenge sets to see if the high performing shuffles of very variable seeds can be trusted in all settings.

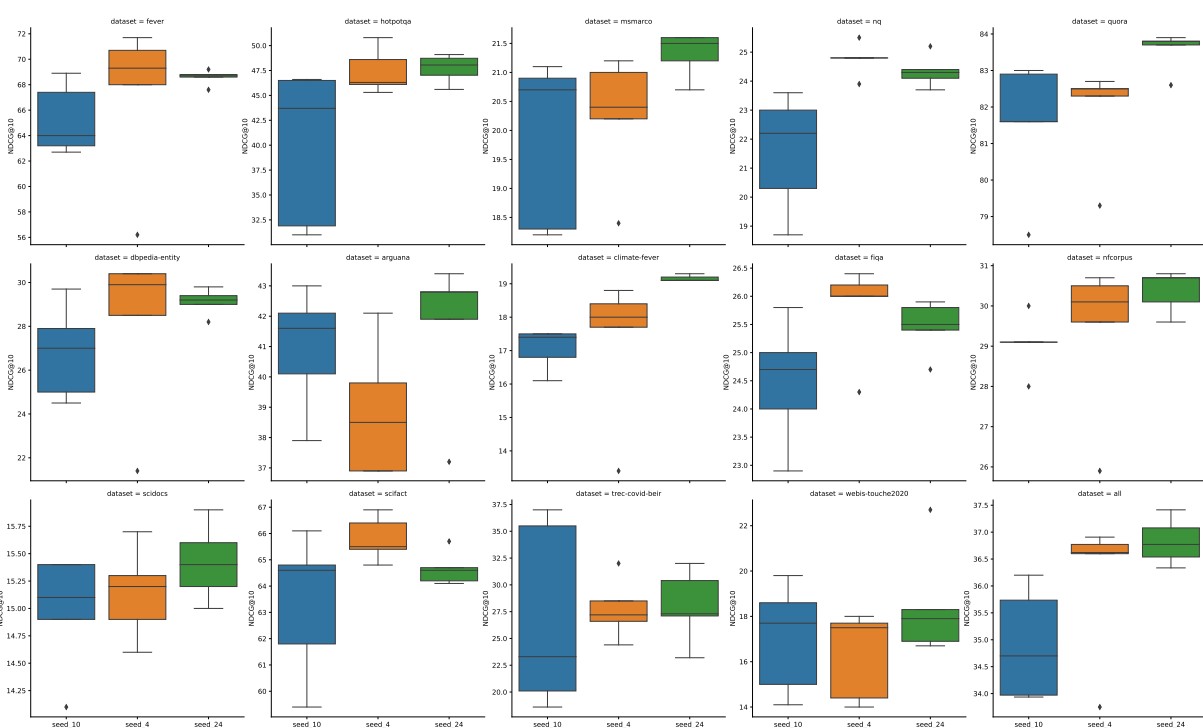

Figure 7: Performance for 5 random datashuffles for a fixed MultiBERT seed - the worst, the best, and a middling seed based on previous experiments. This answers the question of how much variance comes from the random initialisation of parameters, and how much from the data shuffle. It also answers the practical question of 'if you are fine-tuning one model, are you doomed based on the state of the initial model?' The answer is, sort of, but not entirely.

## D Seed 13

MultiContrievers were trained with seeds 0-24 based on respective MultiBerts 0-24. Seed 13 was excluded from all analysis as it displayed repeatedly anomalous behaviour. During the course of contriever training it appeared indistinguishable from other seeds, loss curves looked normal, there were no signs of overfitting. Performance converged to the same level as other MultiContrievers. However, when applied to the datasets of the BEIR benchmark it did not perform at all, with NDCG of between 2 and 20 on each dataset. We retrained once to replicate the behaviour, and then twice more with different seeds for data shuffle, with identical results. We thus exclude it from all analysis. To aid in future investigations we include our initial analysis of seed 13 irregularities here. We follow the method of analysis of representation spaces from Ethayarajh (2019). We measure the L2 norm of all representations in the BiasinBios dataset (272k) as well as average self-similarity of 1000 randomly sampled representations of those bigraphies, as measured by cosine similarity and by dot product. The former answers the question of how much volume the representations occupy, the latter describes the vector space via how conical (anisotropic) or spherical it is.

In Figure 8, we observe that the vector space of MultiContriever 13 is both larger volume and more obtusely anisotropic (i.e. it occupies a wider cone) than other MultiContrievers. The more obtuse anisotropy originates from MultiBert 13, as can be seen in the high variances for both seeds in cosine similarity. But the larger relative volume happens during the training of the MultiContriever and is unique to it. For MultiBert 13, L2 norm is within normal range, and the anomalous seeds are seeds 10 and 23, which both have larger norms and 5x the variance of other seeds. MultiContriever 13, however, has 1.5x the average norms of all other seeds (which have regularised and become closer in values) and 6x the variance of others. Both MultiBert 13 and MultiContriever 13 have very high variance to average cosine similarity, where the effective range of MultiContriever 13 is -0.03 to 0.53, and MultiBert 13 is 0.02 to 0.58, as compared to other models have a range of 0.28-0.32, for both types of models.

We hypothesise that this reveals a limitation of reliance on the dot product for retrieval, any operation reliant on the dot product loses information when there is a chance of a cosine similarity of zero. We leave other investigation – such as why this would persist from a difference of only random seed initialisation, or why this issue would appear in retrieval, but not in any tasks in the MultiBerts paper, or in the contrastive training process – to future work.

We also note that seed 10 was anomalous in performance compared to the other seeds on the BEIR benchmark; not so anomalous as to be excluded, but it was reliably performing poorly. We can see the higher variance in L2 norms for 10 and 23 in MultiBerts, and then for 10 still in MultiContriever (though nothing noticeable in cosine similarity). Seeds 10 and 13 were not found to be anomalous by Sellam et al. (2022), but they did find seed 23 to display strange behaviour and be extremely unbiased (or even anti-biased) on the Winogender benchmark.

We hope that future work will use our models and continue this line of analysis.

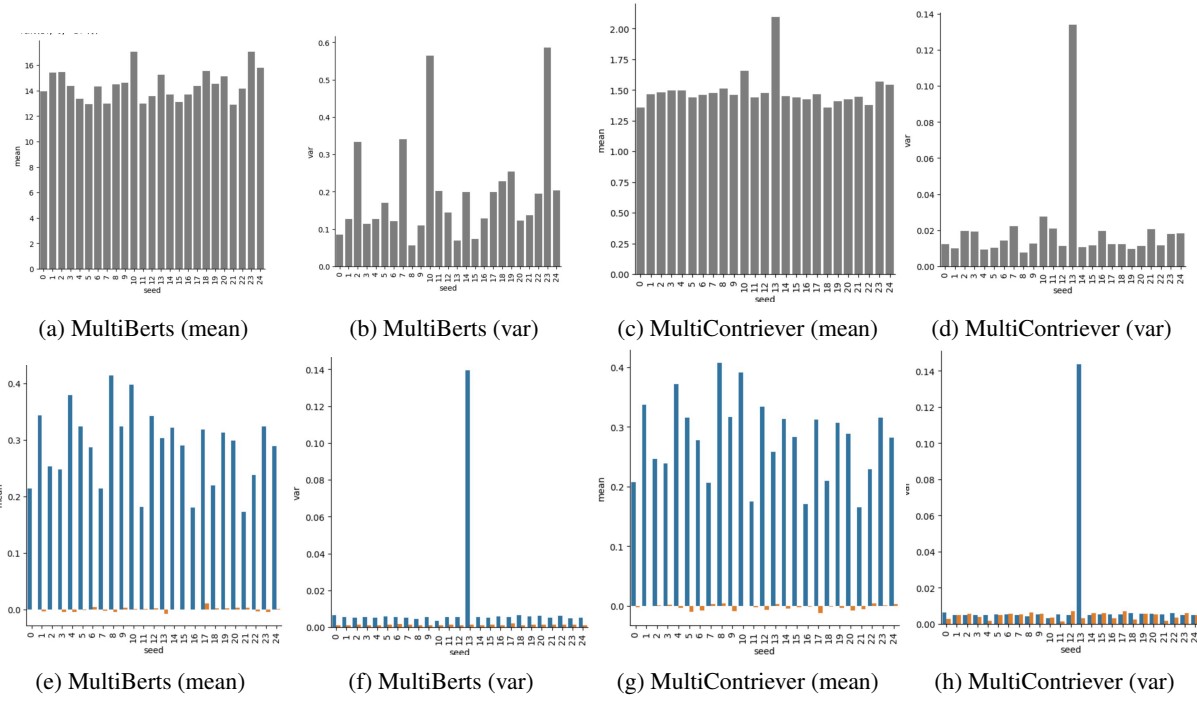

| (a) MultiBerts (mean) | (b) MultiBerts (var) | (c) MultiContriever (mean) | (d) MultiContriever (var) |

| (e) MultiBerts (mean) | (f) MultiBerts (var) | (g) MultiContriever (mean) | (h) MultiContriever (var) |

Figure 8: Top row: mean and var of L2 norms of the full BiasinBios dataset for all MultiBert and MultiContriever seeds. Bottom row: mean and var cosine similarity between 1000 random biographies.

# E Full set of results for correlation between extractability and performance

Full set of correlations between gender compression and performance in Figure 9 and between profession compression and performance in Figure 10. The latter (profession correlation) have misleading regression lines as only three of 24 models had large differences in compression, such that the line is based off insufficient datapoints. It is included for completeness but left out of analysis for that reason. Gender compression numbers (Figure 9) are distributed more evenly. There are four statistically significant correlations (referred to as by row 1-4, and column a-d, such that the upper left cell is 1a and the lower right cell is 4d). Arguana (1a), Scifact (2b), Webis-Touche (3a), and NQ (4b). All have middling correlation coefficients: Arguana -0.41, Scifact 0.41, Webis-Touche 0.31, NQ 0.42. There is also little in common between these datasets, Arguana and Webis-Touche are argumentation, Scifact is fact-checking, and NQ is google-search style questions. As this leaves most datasets with no correlations, we consider the correlation overall to be weak. We do note that the temporal generalisation datasets are overrepresented in this set (Webis-Touche and Scifact), but leave an investigation of that for future work.

Arguana in particular is unique in having a significant *negative* correlation. We have no answers as to why this might be. It may be a fluke due to peculiarities of this dataset: the dataset is small (less thank 2k datapoints), and is not structured in the same way with query (input) and passage (retrieved) but instead uses a full document passage as the query. It is unclear why this might cause a deterioration in performance from better gender or profession encoding (as we observe the same in profession compression). The Arguana task should match the unsupervised training much more closely since they both are matching the relevance of to document chunks. We leave an investigation into the peculiarities of that dataset also to future work.

# F Additional metrics

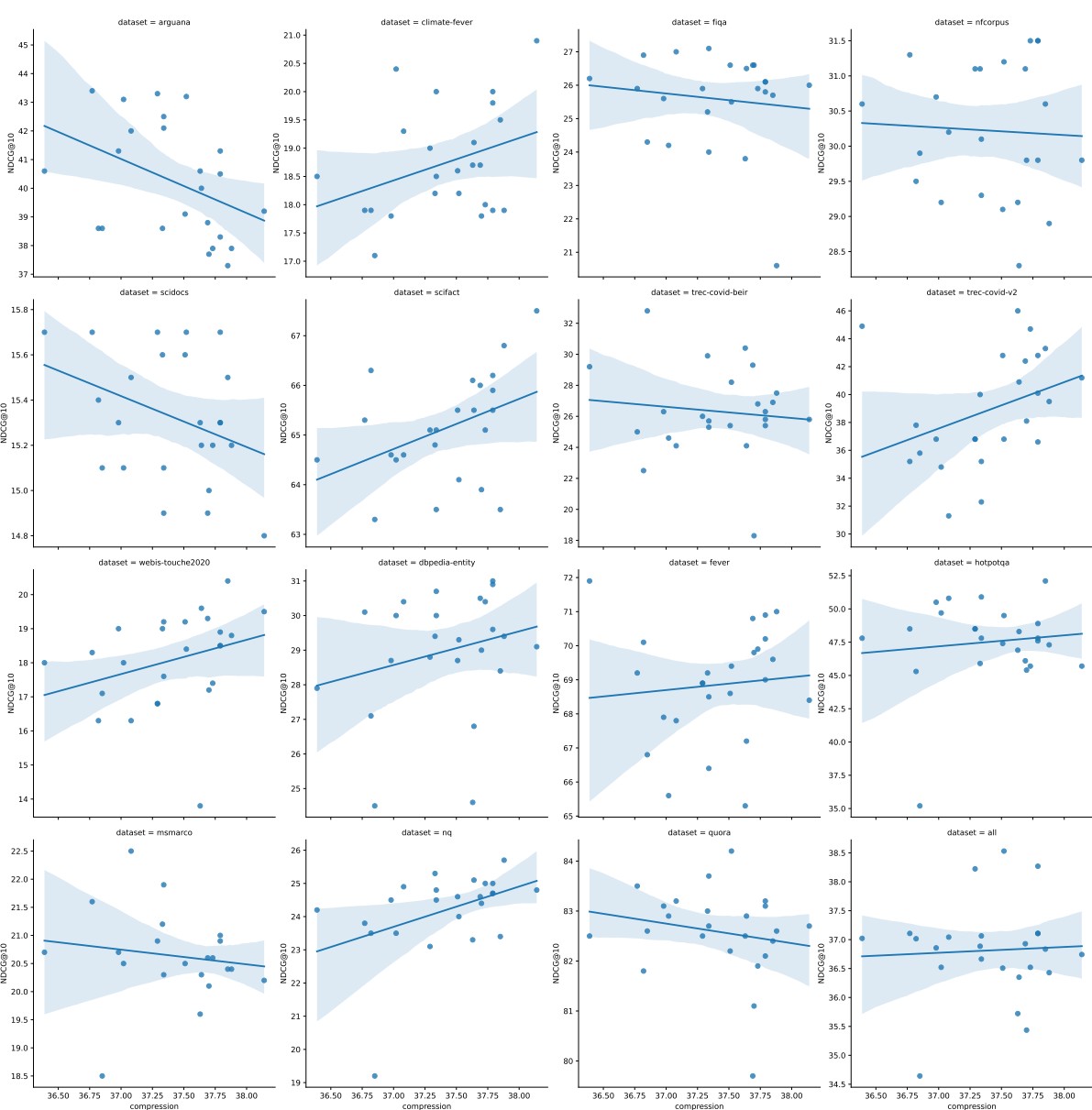

Figure 9: Full set of scatterplots of the correlation between x-axis **gender** compression (ratio of uniform to online codelength) and y-axis performance (NDCG@10), for all datasets individually, and for the average of all BEIR datasets (lower-right). Shaded region is 95% confidence interval.

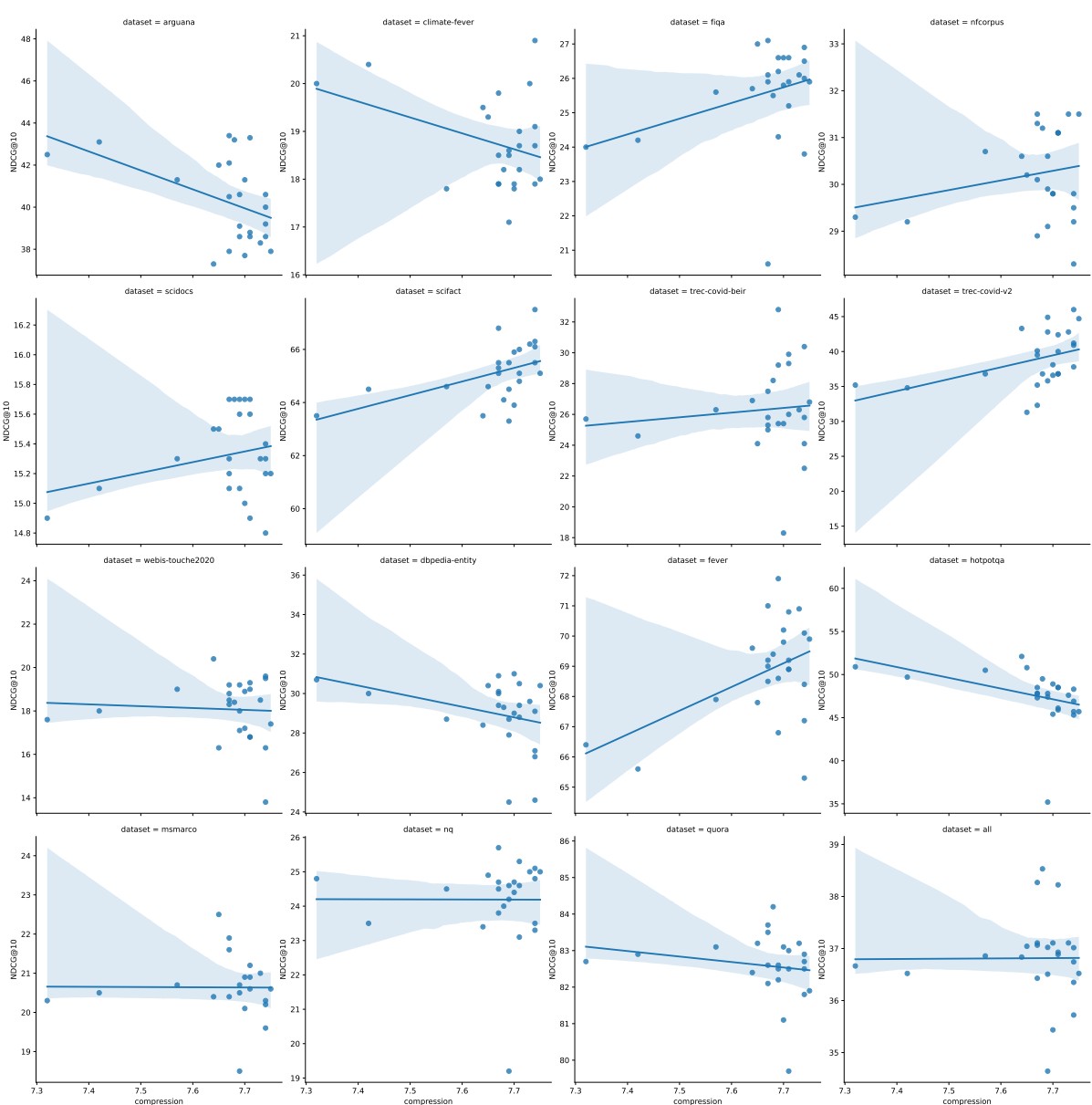

Figure 10: Full set of scatterplots of the correlation between x-axis **profession** compression (ratio of uniform to online codelength) and y-axis performance (NDCG@10), for all datasets individually, and for the average of all BEIR datasets (lower-right). Shaded region is 95% confidence interval.

| | |
|---|---|
| sampling coefficient | 0 |
| pooling | average |
| augmentation | delete |
| probability_augmentation | 0.1 |
| momentum | 0.9995 |
| temperature | 0.05 |
| queue_size | 131072 |
| chunk_length | 256 |
| warmup_steps | 20000 |
| total_steps | 500000 |
| learning_rate | 0.00005 |
| scheduler | linear |
| optimizer | adamw |
| batch_size (per gpu) | 64 |

Table 1: Hyperparameters used for training MultiCon-
trievers.

## G Contriever Training

Each MultiContriever model was initialised from a MultiBert checkpoint for each of the 25 seeds from 0 - 24, accessed at https://huggingface.co/google/multiberts-seed_X where X is an integer from 0 - 24. NB: MultiBerts released many checkpoints to enable study of training dynamics, we use only the final complete checkpoint.

Hyperparameters and training regime is exactly matched to the original Contriever work of (Izacard et al., 2022). Hyperparams can be found in Table 1. Data used was identical to in (Izacard et al., 2022) (from 2019) and was a 50/50 CCNet Wikipedia split.

Each MultiContriever was trained across 4 nodes with 8 GPUs per node (32 GPUs total) for on average 2.5 days. Each MultiContriever was trained for the full 500,000 steps, and checkpointed often; but in all but one seed the best performing checkpoint was the final one (so for that one we use the model at 450,000 steps). This is excepting seed 13, which was anomalous in many other ways (see D).

All MultiContrievers have similar loss and accuracy curves, with seeds 12 and 13 excerpted in Figure 11. All models steeply increase accuracy/decrease loss within 10,000 steps, and then asymptotically approach 69% accuracy by 50,000 steps.

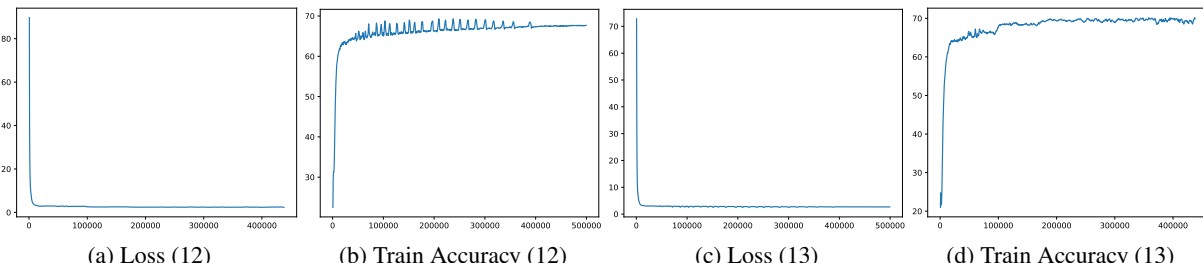

| (a) Loss (12) | (b) Train Accuracy (12) | (c) Loss (13) | (d) Train Accuracy (13) |

Figure 11: Loss and accuracy for seeds 12 and 13, steps on x-axis and loss or accuracy on y-axis.