# OpenReview forum: "MultiContrievers: Analysis of Dense Retrieval Representations"
_EMNLP/2024/Workshop/BlackBoxNLP — BlackboxNLP 2024_

### Official Review · Reviewer_zKjE · 2024-09-08

**Overall Assessment:** 5
**Confidence:** 5

**Best Paper:**

2

**Best Paper Justification:**

Good depth of experiments

**Comments Questions Suggestions And Typos:**

N/A

**Paper Summary:**

The paper presents MultiContrievers, which have been initialize from different MultiBert checkpoints, to understand how specific information such as gender, occupation is encoded in dense representations and how well can they be extracted. It gives unique insights into the blackbox nature of understanding dense representations.

**Summary Of Strengths:**

The paper is well written and easy to understand. Experiments have good depth. Unique insights for understanding how information is encoded in dense representations of retrievers.

**Summary Of Weaknesses:**

Experiments could be focused on few other categories other than gender and occupation to give a more concrete assessment.

---

### Official Review · Reviewer_cRCA · 2024-09-08

**Overall Assessment:** 4
**Confidence:** 3

**Best Paper:**

1

**Best Paper Justification:**

None

**Comments Questions Suggestions And Typos:**

- Figure 4 is too small. Not easy to ready

**Paper Summary:**

This work compares the information captured by dense retrievers system and the language model they are based on. This work uses 25 different checkpoints from MultiBert as the initialization to train MulitContrievers.   This work utilize the minimum description length as the porbing tool to analyze these two models.
Specifically, this work studies the extractability of gender and occupation. As expected, Multicontrievers show higher extractability than MultiBert.
This work also finds that Multicontriever is very senstitve to random initialisation. Even if the model coverges to the same accuracy, its performances on different tasks are dramastically different.
Surprisingly, this work shows that the task  performance (NQ, MSMARCO) are not correlated with the extractability.

**Summary Of Strengths:**

- First work to analyze the dense retrieve vector reprentations.
- Extensive discussion and analysis of the experimental results.

**Summary Of Weaknesses:**

None

---

### Official Review · Reviewer_roMP · 2024-09-10

**Overall Assessment:** 3
**Confidence:** 3

**Best Paper:**

1

**Best Paper Justification:**

NA

**Comments Questions Suggestions And Typos:**

- Figure 4 is hard to read, I would recommend highlighting a few datasets and moving the rest to Appendix.
- Any reason for restricting your analysis to just gender and occupation?
- The definition for 'allocational fairness' is not immediately clear to me (line 268).

**Paper Summary:**

This paper studies the effect of random initialization for the dense retrieval task. They build upon prior work on BERT, specifically, MultiBERT that provided 25 checkpoints of BERT with different weight initialization and data shuffling. MultiContrievers (this work) additionally include random initialization of data shuffling for contrastive learning. The paper analyzes extractability of gender and occupation, as well as the effect of downstream performance on BEIR benchmark.

Overall, the paper is well written and provides insights into the effects of initialization of dense retrievers. Compared to MultiBERTs, authors find that MultiContrievers showcase better extractability of both gender and occupation. On the BEIR benchmark, MultiContrievers show a wide range of performance on individual datasets, sometimes outperforming a supervised model. Authors highlight that while gender extractability is higher with MultiContrievers, their impact on a diverse benchmark such as BEIR is minimal.

**Summary Of Strengths:**

- Well written, with clear focus on the research questions and experiments to support the core arguments of the paper. The MultiContriever checkpoints could be useful to future work on interpretability.
- The study about the necessity of gender information on the NQ-gender and overall NQ is interesting.
- The wide range of performance of MultiContriever checkpoints on individual datasets in BEIR is surprising. I think this need more study in the future work (also related, see my comment in weaknesses).

**Summary Of Weaknesses:**

- It's not immediately clear if this analysis will hold for recent dense retrieval models. While Contriever (Izacard et al., 2022) is a reasonable baseline, there are a number of dense retrieval models released in the last two years. Will the initialization (and shuffling) have the same effect on current state-of-the-art retrievers? Or is it possible that its only Contriever that's more sensitive to the initialization?
- Given that authors find MultiContriever performance to very sensitive to random seed, it would be useful to present directions for future work on dense retrieval models.

---

### Decision · Program_Chairs · 2024-09-19

**Decision:**

Accept

**Comment:**

Reviewers appreciated the analysis of dense retrievers and resulting insights. One concern is that Contriever is relatively old compared to new dense retrievers. It would be great to see an analysis of a newer model in the camera ready version, but the paper is already interesting as is.